# Ectopic Expression of *HbRPW8-a* from *Hevea brasiliensis* Improves *Arabidopsis thaliana* Resistance to Powdery Mildew Fungi (*Erysiphe cichoracearum* UCSC1)

**DOI:** 10.3390/ijms232012588

**Published:** 2022-10-20

**Authors:** Xiaoli Li, Qiguang He, Yuhan Liu, Xinze Xu, Qingbiao Xie, Zhigang Li, Chunhua Lin, Wenbo Liu, Daipeng Chen, Xiao Li, Weiguo Miao

**Affiliations:** 1School of Plant Protection/Key Laboratory of Green Prevention and Control of Tropical Plant Diseases and Pests, Ministry of Education, Hainan University, Haikou 570228, China; 2Hainan Provincial Key Laboratory of Tropical Crops Cultivation and Physiology, Key Laboratory of Biology and Genetic Resources of Rubber Tree, Ministry of Agriculture and Rural Affairs, Rubber Research Institute, Chinese Academy of Tropical Agricultural Sciences, Haikou 571101, China; 3Hainan Key Laboratory for Sustainable Utilization of Tropical Bioresources, Institute of Tropical Crops, Hainan University, Haikou 570228, China

**Keywords:** *Hevea brasiliensis*, *RPW8*, powdery mildew, resistance gene, salicylic acid

## Abstract

The *RPW8s* (Resistance to Powdery Mildew 8) are atypical broad-spectrum resistance genes that provide resistance to the powdery mildew fungi. Powdery mildew of rubber tree is one of the serious fungal diseases that affect tree growth and latex production. However, the *RPW8* homologs in rubber tree and their role of resistance to powdery mildew remain unclear. In this study, four *RPW8* genes, HbRPW8-a, b, c, d, were identified in rubber tree, and phylogenetic analysis showed that HbRPW8-a was clustered with AtRPW8.1 and AtRPW8.2 of *Arabidopsis.* The HbRPW8-a protein was localized on the plasma membrane and its expression in rubber tree was significantly induced upon powdery mildew infection. Transient expression of *HbRPW8-a* in tobacco leaves induced plant immune responses, including the accumulation of reactive oxygen species and the deposition of callose in plant cells, which was similar to that induced by *AtRPW8.2*. Consistently, overexpression of *HbRPW8-a* in *Arabidopsis thaliana* enhanced plant resistance to *Erysiphe cichoracearum* UCSC1 and *Pseudomonas syringae* pv. *tomato* DC30000 (*Pst*DC3000). Moreover, such *HbRPW8-a* mediated resistance to powdery mildew was in a salicylic acid (SA) dependent manner. Taken together, we demonstrated a new RPW8 member in rubber tree, HbRPW8-a, which could potentially contribute the resistance to powdery mildew.

## 1. Introduction

Plants have developed two unique pathways to response for pathogen attacks, namely pathogen-associated molecular pattern-triggered immunity (PTI) and effector-triggered immunity (ETI), which work together to protect plants [1,2,3]. The plant immune responses usually involve the burst of reactive oxygen species (ROS), activation of mitogen-activated protein kinase (MAPK) signaling and downstream PATHOGENESIS-RELATED (PR) genes, regulation of phytohormone signaling pathways, and induction of hypersensitive response (HR) [4,5,6,7,8,9]. The ROS burst is one of the early cellular responses in plant cells, serving as a toxin barrier against subsequent pathogen infection [10]. Moreover, the ROS burst can also strengthen the plant cell walls through oxidative cross-link of polymers to inhibit pathogen growth [11,12,13]. While HR is a rapid cell death phenomenon induced by pathogen or their metabolites, occurring at the point of pathogen penetration to prevent further pathogen spreading [14,15,16,17,18,19,20]. The plant immune system is also regulated by plant hormones, especially jasmonic acid (JA) and SA, which play an important role in basic resistance against various plant pathogens. Furthermore, the phytohormone signaling pathways are thought to be a cost-effective manner for plant to coordinate biotic stress responses with growth and survival [21,22,23,24]. In plants, the recognition of pathogen avirulent (Avr) proteins by the resistance (R) proteins is an effective defense mechanism against pathogen infection [25,26,27,28]. *AtRPW8.1* and *AtRPW8.2* genes were conferred a broad-spectrum resistance to *Erysiphe* spp. in *Arabidopsis* Wa and Ms [29,30]. As atypical R proteins, they contain only the RPW8 domain, and lack of the nucleotide-binding site and leucine-rich repeats (NB-LRR) domain that appears in most of R proteins [31,32,33,34,35]. Although both *AtRPW8.1* and *AtRPW8.2* can mediate plant resistance through the SA and Enhanced Disease Susceptibility 1 (EDS1) signaling pathways [36], their underlying mechanisms seem different. *AtRPW8.1* stimulated H_2_O_2_ accumulation in chloroplasts to enhance PTI, while *AtRPW8.2* stimulated high concentration of H_2_O_2_ in the haustorium complex to achieve the disease resistance [37,38,39]. So far, various RPW8 proteins with positive effects on plant resistance have been identified in different plants [40,41,42,43].

The rubber tree (*Hevea brasiliensis* Muell. Arg.) is mainly distributed in areas within 10° north-south latitude. Its production of latex is an important industrial raw material, which cannot be replaced by artificial synthesis [44]. However, both rubber tree growth and its latex production have been affected by biotic and abiotic stresses [45,46,47,48]. The powdery mildew caused by *Erysiphe quercicola* is one of the serious diseases among the biotic stresses. This disease can affect immature tissues such as young leaves, buds and inflorescences, leading to reduced latex yield of rubber trees [49,50,51]. At present, sulfur powder and magnetic levitation agents are mainly used to prevent powdery mildew in rubber trees [49]. However, the use of these agents has the disadvantages of poor adhesion and environmental pollution, and is not a long-term solution for powdery mildew resistance. The discovery of plant *R* genes (including their homologs and analogs) opened up new possibilities for managing plant diseases that are caused by pathogens; the recent advances in plant genome sequencing, allows for the identification of the novel *R* genes. To date, several powdery mildew resistant/susceptible genes of rubber trees have been reported, including *HbSGT1a*, *HbSGT1b* and *HbHSP90.1. HbSGT1a* and *HbSGT1b* were powdery mildew resistance-related genes. Moreover, *HbMlo12* is a candidate gene conferring powdery mildew susceptibility [52,53]. However, the mechanism underlying *E. quercicola* and *H. brasiliensis* interaction and the defense-related genes in rubber tree remains elusive. Furthermore, compared with more than 80 powdery mildew resistance genes in wheat, the number of reported powdery mildew resistance genes in rubber trees was limited. Additionally, due to the rapid evolution of pathogen, *R*-genes are often temporally effective for specific pathogens [54,55,56,57]. Therefore, exploring resistance genes with a broad-spectrum resistance against pathogen may play an important role for screening new cultivated rubber tree varieties while retaining high-quality latex production.

Here, we identified four *RPW8* genes, *HbRPW8-a*, *b*, *c*, *d*, from the powdery mildew mildly susceptible rubber tree cultivar Reyan 7-33-97 and analyzed their expression patterns during the period of *E. quercicola* infection. Of these genes, *HbRPW8-a* clustered with AtRPW8.1 and AtRPW8.2 of *Arabidopsis* was selected for further study. By transient expression in tobacco leaves and overexpression in *Arabidopsis* plants revealed HbRPW8-a induced plant immune responses associated with plant defense. Finally, we showed that HbRPW8-a mediated resistance in *Arabidopsis* was dependent on the SA signaling.

## 2. Results

### 2.1. Isolation and Identification of HbRPW8s

To identify *RPW8* genes in rubber tree, AtRPW8.1 and AtRPW8.2 protein full-length sequence was used to blast against the genome of *H. brasiliensis*. Four *HbRPW8* genes were obtained, and designated as *HbRPW8-a*, *HbRPW8-b*, *HbRPW8-c* and *HbRPW8-d*, respectively. Compared with the other three HbRPW8s (about 800 amino acids), HbRPW8-a was much smaller, with only 189 amino acids (Appendix A).

Protein domain analysis showed that HbRPW8-a only contained the RPW8 domain, whereas other HbRPW8s contained RPW8, NB-ARC and LRR domains (Figure 1a). As expected, phylogenetic analysis of HbRPW8s, AtRPW8s and VpRPW8s showed that HbRPW8-a, AtRPW8.1 and AtRPW8.2 were clustered together (Figure 1b). 

### 2.2. Expression of HbRPW8s upon E. Quercicola Infection

To investigate the possible roles of HbRPW8s in powdery mildew disease, the transcriptional expression levels of *HbRPW8s* gene were measured at different times after *E. quercicola* infection (Figure 2). The expression patterns of *HbRPW8-b* and *c* were similar, with a strong reduction at 48 hours post infection (hpi) (the second penetration process of *E. quercicola*), while the expression of HbRPW8-d was relatively stable during the whole infection period.

Interestingly, the expression level of *HbRPW8-a* was significantly induced at 13 hpi (primary infection period of *E. quercicola*) and 24 hpi (the haustorium development stage during *E. quercicola* infection) compared to that of control. Combined with the high similarity between HbRPW8-a and AtRPW8.1/8.2, HbRPW8-a was selected for further investigate in plant defense.

### 2.3. Transient Expression of HbRPW8-a Induces Plant Immune Responses in N. benthamiana Leaves

AtRPW8.1 and AtRPW8.2 were reported to be localized in the endomembrane system to target the haustorium upon powdery mildew infection. To determine the subcellular localization of HbRPW8-a, the HbRPW8-a-mScarlet and RLK-GFP (a plasma membrane control) fusion proteins were transiently expressed in *N. benthamiana* leaves. Using confocal microscopy, we found that the fluorescence signal of HbRPW8-a-mScarlet fusion protein was co-localized with RLK-GFP at the plasma membrane (Figure 3), indicating HbRPW8-a is a membrane localized protein.

In addition to subcellular localization of HbRPW8-a, we also observed HbRPW8-a induced callose deposition in *N. benthamiana* leaves, similar to that induced by AtRPW8.2 (Figure 4a,b). Additionally, similar to that of AtRPW8.2-GFP, the HbRPW8-GFP fusion protein can also induce ROS burst in tobacco cells (Figure 4c,d). These results suggest that HbRPW8-a, like AtRPW8.2, could induce plant immune responses.

### 2.4. Over-expressing HbRPW8-a in Arabidopsis Col Enhances Resistance to Powdery Mildew

Different from the *Arabidopsis* Wa and Ms ecotypes containing *AtRPW8.1*/*AtRPW8.2* with resistance to powdery mildew, the *Arabidopsis* Col is susceptible to powdery mildew due to the lack of *AtRPW8.1*/*AtRPW8.2*. To further investigate the role of HbRPW8-a in resistance to powdery mildew, we overexpressed *HbRPW8-a* in the *Arabidopsis* Col plants and obtained T3 transgenic homozygous lines (Appendix A). The high expression level of *HbRPW8-a* gene in these lines was confirmed by qRT-PCR (Appendix A).

Three T3 *HbRPW8-a* overexpression (#8, #16 and #27), Col-WT, Wa-WT and the *Arabidopsis* mutant *pad4* at Col background (a susceptible mutant to powdery mildew), were inoculated with powdery mildew (*E. cichoracearum* UCSC1), respectively. After 7 days of inoculation (dpi), the *pad4* displayed severe disease, with the entire leaves covered by powdery mildew fungi hyphae. Col-WT plants also had visible disease symptoms, with large areas of hyphae on the fully expanded leaves. While the *HbRPW8-a* overexpression lines (#8, #16 and #27) showed limited disease symptoms, with fewer hyphae on the leaves, and the resistant plant Wa-WT was basically disease free (Figure 5a). Additionally, we measured the basal immune responses in *Arabidopsis* plants after powdery mildew inoculation. Consistent with their disease symptoms, the callose deposition and ROS accumulation in leaves of *HbRPW8-a* overexpression lines were significantly higher than those in Col-WT and *pad4* mutant (Figure 5b–e). By trypan blue, we observed that blue-stained necrotic cells were abundant in the leaves of *HbRPW8-a* overexpression lines, but only at the leaf margins in Col-WT (Figure 5c). We also counted the number of conidiophores per colony from these plants and revealed that the *HbRPW8-a* overexpression lines had a significantly fewer conidiophores than that of Col-WT (Figure 5f). Similar to that in *AtRPW8.1*-expressed *Arabidopsis* plants [39], the epidermal cells of *HbRPW8-a* overexpression lines were enriched in hydrogen peroxide surrounding germinating conidia (Figure 5g). These results reveal that overexpression of *HbRPW8-a* can induce the basal immune response in *Arabidopsis* Col and prevent powdery mildew colonization effectively.

### 2.5. Over-expressing HbRPW8-a in Arabidopsis Col Enhanced Plant Resistance to Bacteria

Flagellin 22 (flg22), as a conserved molecule pathogen-associated pattern, can trigger innate immune responses in plants [58]. After flg22 treatment, the accumulation of ROS and callose deposition in Col-WT leaves were continuously increased from 24 h to 72 h, and these responses were even stronger in the *HbRPW8-a* overexpression lines (Figure 6a–d).

To test whether HbRPW8-a could play a role in bacterial infection, we treated the *HbRPW8-a* overexpression lines with virulent *Pst*DC3000. Col-WT plants displayed severe chlorosis in leaves with enhanced bacterial growth, whereas *HbRPW8-a* overexpression lines showed only mild disease symptoms with lower bacterial numbers (Figure 6e–f). These results indicate that the expression of *HbRPW8-a* could also enhance resistance to biotrophic bacteria *P. syringae* and inhibit the colonization of *Pst*DC3000 in *Arabidopsis* Col plants. 

### 2.6. Over-Expression of HbRPW8-a Cannot Improve Plant Resistance to Phytophthora Capsici and Botrytis Cinerea

Pathogen-induced cell death is important in plant resistance to biotrophic pathogens [59]. We used *B. cinereal* and *P. capsici* to investigate whether HbRPW8-a could also be involved in plant resistance to necrotrophic pathogens. The lesion diameters of *B. cinerea* spore infected leaves were measured at 3 dpi, however, no significant difference was observed between *HbRPW8-a* overexpression lines and Col-WT (Figure 7a,b).

*P. capsici* spores were inoculated on tobacco leaves that transiently expressed GFP, HbRPW8-a-GFP or AtRPW8.2-GFP protein after 24 h. Similarly, the lesions of the infected sites were indistinguishable between control and HbRPW8-a-GFP/ AtRPW8.2-GFP expressed regions at 3 dpi (Figure 7c,d). These results indicate that HbRPW8-a has no contribution to plant resistance to *P. capsici* and *B. cinerea*.

### 2.7. SA Is Required for HbRPW8-a-Mediated Powdery Mildew Resistance

SA, as one of the hormone signaling pathways, plays an important role in plant immune responses. Expressing a bacterial salicylate hydroxylase (encoded by *NahG*) in *Arabidopsis* plants was shown to convert SA to catechol [60], causing a decreased disease-resistance response due low accumulated SA [61].

To examine whether HbRPW8-a mediated plant resistance is dependent on SA signaling, we overexpressed *HbRPW8-a* in the *NahG* plants at Col background, and selected T3 transgenic homologous lines #1, #5 and #13 for further study (Appendix A).

The disease symptom analysis in Figure 8 showed that the overexpression of *HbRPW8-a* in SA-deficient transgenic *NahG* plants failed to be resistant to powdery mildew, with similar numbers of hyphae distributed on the leaves compared to that of Col-WT (Figure 8a,g). Interestingly, the *NahG* plants and its HbRPW8-a transgenic lines had significantly higher ROS accumulation but less callose deposition than that of Col-WT, which caused a larger area of cell death at 3 dpi (Figure 8b–f), suggesting that SA signaling may contribute to the HbRPW8-a mediated plant resistance via regulating callose deposition.

## 3. Discussion

*RPW8* genes were first reported as an atypical R gene in *Arabidopsis* in 2001 [30]. They encoded proteins containing an N-terminal transmembrane domain and one or two coiled-coil domains, known as the RPW8 domain [30]. In recent years, proteins with conserved RPW8 domains have also been found in *Brassica napus*, *strawberries (Fragaria × ananassa)*, *V. pseudoreticulata*, *Cucurbita pepo* L., *Akebia trifoliata* and *Lagenaria siceraria (Molina) Standl*. (Figure 9); (for sequence information, see Appendix A) [40,41,42,62,63,64]. Among them, several RPW8s have been proved to be resistant to powdery mildew. In this study, we identified four HbRPW8s with the conserved RPW8 domain in rubber tree (Figure 1). By phylogenetic analysis, we showed that HbRPW8-a together with AtRPW8.1 and AtRPW8.2 were grouped into the same clade, indicating that HbRPW8-a protein was similar to that of AtRPW8.1 and AtRPW8.2 (Figure 1B). Similar to that of *Lsi04g015960* (encoded RPW8-NBS-LRR protein) and *Cp4.1LG10g02780* (encoded RPW8-NBS-LRR protein), *HbRPW8-a* was strongly induced by powdery mildew at 13 and 24 hpi, which corresponds to the early stages of the interaction between powdery mildew and host plants, suggesting the role of *HbRPW8-a* in rubber trees in response to powdery mildew.

*AtRPW8* was reported as a broad-spectrum resistance gene [39,40], expression of *AtRPW8* leads to enhanced resistance to four *Arabidopsis* powdery mildew species, tobacco powdery mildew, *Hyaloperonospora parasitica*, *Cauliflower mosaic virus*, *Pyricularia oryza*, *Xanthomonas oryzae pv. oryzae* and other plant pathogens [30,65]. Homogeneously, the *VpRPW8s* could also enhance plant resistance to *P. viticola*, *P. capsici*, and *P. parasitica* [41]. Additionally, *MdRNL2* and *MdRNL6* (encode RPW8-NBS-LRR protein) were considered to be spectrum fungal resistance genes [66]. By transient expression in tobacco leaves and overexpression in Col-WT *Arabidopsis*, we showed that *HbRPW8-a* is a broad-spectrum resistance gene with resistance to *E. cichoracearum UCSC1* and *Pst*DC3000. Overexpression of *AtRPW8.1* and *AtRPW8.2* can active oxygen burst and callose deposition and induce plant HR response, which promotes neoplastic pathogens *Alternaria* and *Botrytis* spp. infection. In contrast, overexpressing of *HbRPW8-a* could not increase the sensitivity of plants to neoplastic pathogens *B.cinerea* and *P. capsici*. However, further study is required to understand the different roles between HbRPW8-a and AtRPW8.1/AtRPW8.2.

Plant RPW8s are diverse in terms of subcellular localization [41]. Both AtRPW8.1:YFP and AtRPW8.2:YFP appeared to be localized mainly in the endomembrane system [65], especially for AtRPW8.2:YFP, targeted to haustorium when powdery mildew infected, which is key to activation of resistance at the host-pathogen interface. BnHRa YFP and BnHRb-YFP were mainly localized to the extra-haustorial membrane [40]. VpRPW8s mainly located in the cytoplasm [41]. Interestingly, we found that HbRPW8-a was plasma membrane localized protein (Figure 3). AtADR1(RPW8-NBS-LRR protein), an *Arabidopsis* HeLo-/RPW8-like domain ‘helper’ NLR (RNL), was also localized on plasma membrane to induce immune signaling and cell death [67,68,69,70]. It will be interesting to investigate whether the plasma membrane-localization of HbRPW8-a is required for its induction of innate immune signaling. 

Previous study showed that AtRPW8-mediated resistance to powdery mildew in Arabidopsis was dependent on SA signaling [71]. Similarly, we found that decreased SA in NahG plants blocked the function of HbRPW8-a in inhibiting fungal growth. Interestingly, ROS was significantly high accumulated in the leaves of NahG plants, which is probably because SA acts as direct or indirect antioxidants to protect cells from oxidative damage, observed in animal systems [72,73,74]. In addition to SA, it is very possible that other signaling pathways are also involved in HbRPW8-a mediated resistance to powdery mildew in Arabidopsis, which requires further study in the future. 

## 4. Materials and Methods

### 4.1. Biological Materials and Growth Conditions

The *Arabidopsis thaliana ecotype Columbia-*0, *pad4*, *Nicotiana benthamiana*, *and Hevea brasiliensis* (Reyan 7-33-97) were all from our laboratory. The *Arabidopsis thaliana ecotype* Wa was kindly offered by Prof. Wenming Wang’s Lab. The *Arabidopsis* NahG was kindly offered by Chaozu He’s Lab. Rubber tree susceptible cultivar CATAS7-33-97 plants [40] were grown in the experimental plantation of Hainan University. *N. benthamiana*, *A. thaliana* wild type (Col-0 and Wa ecotype), and the *pad4* mutant at Col background were grown in a greenhouse at 24 ± 2 °C under a 16 h light/8 h dark photoperiod. *E. cichoracearum* UCSC1 was cultured on *A. thaliana pad4* mutant plants. *Pst*DC3000 was preserved at −80 °C. *B. cinerea* and *P. capsica* were maintained at 28 °C on Potato Glucose Agar (PDA) in the dark. 

### 4.2. Quantitative Real-Time PCR

For qRT-PCR, rubber tree bronze leaves inoculated with powdery mildew spores were sampled at 0, 4, 8,13, 24, 48, and 72 hpi. The total RNA was extracted by a plant total RNA extraction kit (Tiangen Biotech., Beijing, China). The first-strand cDNA was synthesized by the EasyScript^®^ One-Step gDNA Removal and cDNA Synthesis SuperMix kit (TRAN, Beijing, China). qRT-PCR was then performed with SYBR green (TRAN, Beijing, China) in the LightCycler^®^ 96 System (Roche, Switzerland). *Hb18s* was used as the internal control, and all used primers were listed in Appendix A. Relative expression level was calculated by 2^−ΔΔCt^ method [75], in triplicates for each gene.

### 4.3. Sequence and Phylogenetic Analysis

ProtParam (http://web.expasy.org/protparam/, accessed on 16 December 2019), and NCBI-CDD (http://www.ncbi.nlm.nih.gov/Structure/cdd/wrpsb.cgi, accessed on 20 December 2019), tools were used for searching RPW8s in the genome of rubber tree. 

The phylogeny was inferred based on a multiple sequence alignment of the full-length amino acid sequences of the reported *V. pseudoreticulata* and *A. thaliana* RPW8s (AtRPW8.1/8.2, HR1-HR4). The phylogenetic tree was constructed via MEGA 6.0. The numbers at the end of protein names indicate the accession number of each sequence in the NCBI database.

### 4.4. Subcellular Localization and Transient Expression

For subcellular localization analysis, 35s-HbRPW8-a-mScarlet was constructed using a 35s-mScarlet plasmid. The RLK (At4g23740), a plasma membrane localized protein [76], was constructed as positive control (pcambia1300-35s-RLK-GFP). The recombinant plasmids were introduced into *A.tumefaciens* strain GV3101. *Agrobacterium* containing recombinant plasmids was then injected into *N. benthamiana* epidermal cells. The RLK-GFP and HbRPW8-a-mScarlet fusion proteins were observed after 48 h at 488 nm and 569 nm by confocal microscope (Leica TCS SP8, Germany), respectively [77].

### 4.5. Plant Transformation and Disease Assays

Plant transformation in *Arabidopsis* Col-WT and NahG was performed by the flower dip method [78]. Three T3 homozygous lines of 35s-HbRPW8-a-mScarlet were selected for subsequent experiments.

*P. capsica*, *B. cinerea*, *Pst*DC3000, and *E.cichoracearum* were used for pathogenicity tests. Four-week-old tobacco leaves were inoculated with *A.tumefaciens* containing 35s-HbRPW8-a-GFP/35s-AtRPW8.2-GFP recombinant plasmids. After inoculation for 24 hpi, the tobacco leaves were inoculated with *P. capsici* spore suspension at a concentration of 5 × 10^4^ spores mL^−1^ [79] and were kept under moisture retention in vitro. Five-week-old *Arabidopsis* leaves were inoculated with *B. cinerea* spores at the concentration of 1.5 × 10^6^ spores mL^−1^ [80], which were also kept under moisture retention in vitro. The diameter of necrotic spots was measured after three days.

The transgenic *Arabidopsis* leaves were sprayed by *B. cinerea* spores at the concentration of 2 × 10^5^ spores mL^−1^, and the size of lesion was measured after 3 days of treatment. *Pst*DC3000 treatment with OD = 0.6 was used to observe the leave disease, the OD = 0.1 solution was used to count the bacterial concentration at the time of 0, 24, 48 and 72 hpi. For powdery mildew infection, *E. cichoracearum* grown for about 10 days was inoculated on transgenic lines for disease test [81]. For flg22 treatment, four-week-old *Arabidopsis* plants of Col-WT and transgenic lines were sprayed with 1 μM flg22 and harvested after 24, 48, 72 hpi.

### 4.6. ROS Measurement and Callose Staining

For ROS burst assay, four-week-old tobacco leaves and *Arabidopsis* leaves were collected and were stained with 3,3′-diaminobenzidine (DAB, 2 mg mL^−1^) solution overnight. To investigate ROS accumulation, cleaned leaves were boiled in ethanol (95%) until pigments were removed. Callose staining assay was performed as described previously [82]. In brief, treated leaves were collected and socked in ethanol (95%) overnight to remove the pigments, and then incubated in the staining solution (0.1% aniline blue, 67 mmol/L Na_2_HPO_4_, pH 12.0) for 1 h in dark at room temperature. Callose deposition was observed under an epifluorescence microscope (Carl Zeiss, Oberkochen, Germany) and photographed.

## 5. Conclusions

In conclusion, we have identified four *HbRPW8* genes from rubber tree in this study. By taking advantage of an *Arabidopsis* Col ecotype that lacks AtRPW8.1/8.2, we show that HbRPW8-a, a protein localized on the plasma membrane, has broad resistance to several plant pathogens including powdery mildew. Our study reveals a new member of the plant RPW8 family, providing a foundation for future studies and potential applications in rubber tree variety improvement.

## Figures and Tables

**Figure 1 ijms-23-12588-f001:**
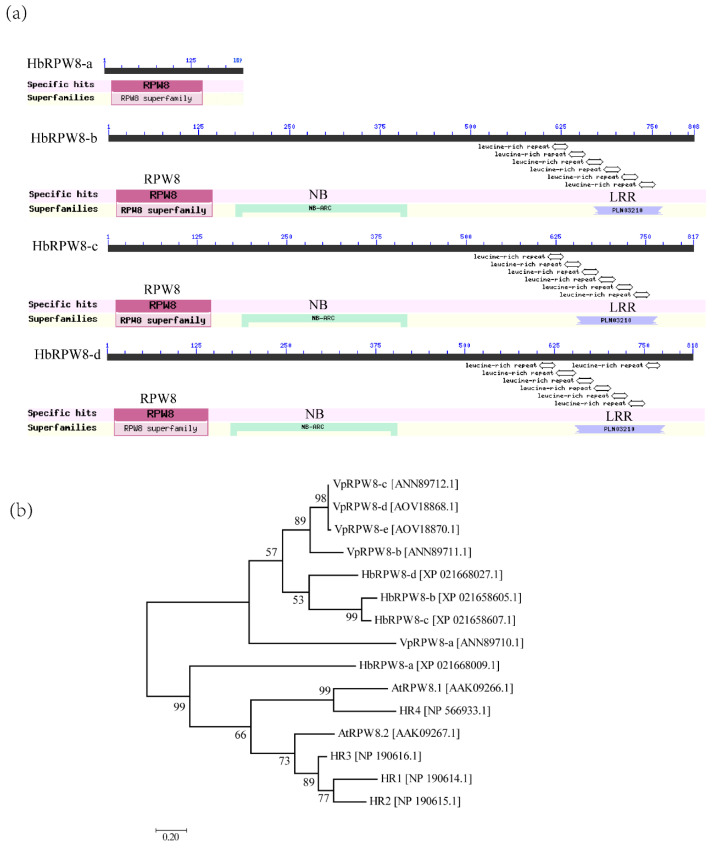
Protein domain and phylogenetic analysis of HbRPW8s. (**a**) Protein domain analysis of HbRPW8s. HbRPW8s contained RPW8 domain, conserved disease resistance domains NB (nucleotide binding sites) and LRR (leucine-rich repeats), except for HbRPW8-a, with only the RPW8 domain (7-128aa); (**b**) phylogenetic analysis of HbRPW8s with other RPW8s in *V. pseudoreticulata* and *A*. *thaliana*.

**Figure 2 ijms-23-12588-f002:**
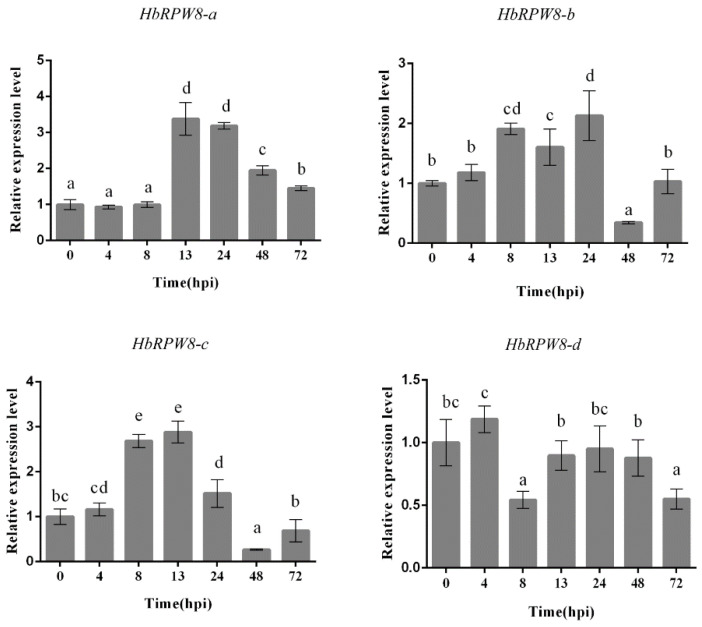
Transcriptional expression of *HbRPW8s* in response to *E. quercicola.* The relative expression of each gene was calibrated against 0 h uninoculated samples. The experiment was performed with three biological replicates, and six plants were used in each independent trial. The data were analyzed by one-way ANOVA and Duncan’s multiple comparison tests. Data were showed as means ± standard error. Bars with different lower-case letters show significant differences at the level of *p* < 0.05.

**Figure 3 ijms-23-12588-f003:**
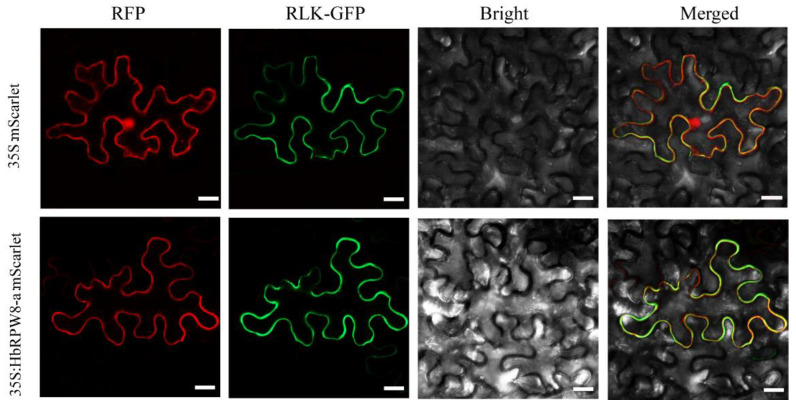
Subcellular localization of HbRPW8-a protein by transient expression in tobacco epidermal cells. The plasma membrane localized AtRLK-GFP was used as positive control. The 35S: mScarlet empty vector was used as negative control. Scale bar = 20 μm.

**Figure 4 ijms-23-12588-f004:**
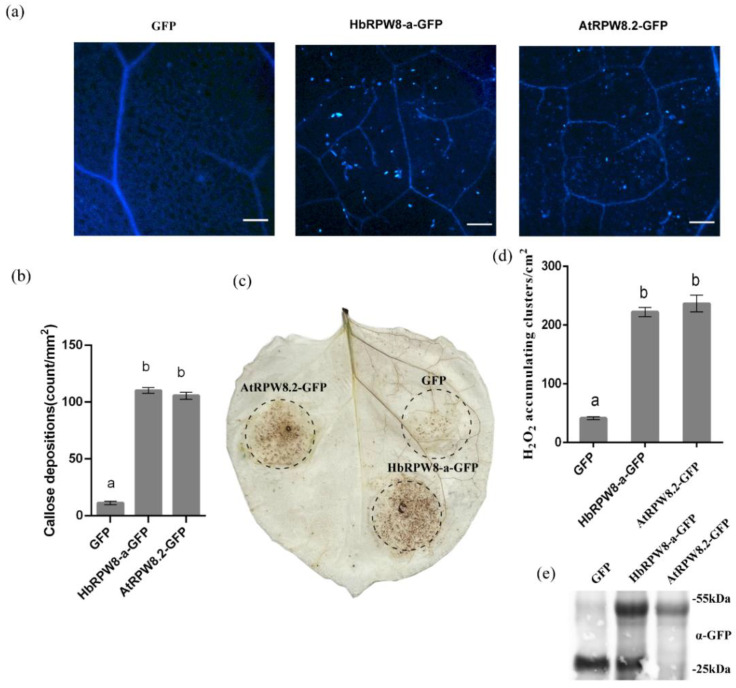
Plant immune responses induced by transient expression of *HbRPW8-a* in *N. benthamiana* leaves. A total of five plants were used for each treatment. Each test contained three biological replicates. (**a**) Callose deposition induced by *HbRPW8-a* and *AtRPW8.2* genes in *N. benthamiana* leaves; scale bar = 100 μm. (**b**) Average numbers of callose deposits per 1 mm^2^; (**c**) ROS accumulation induced by *HbRPW8-a* and *AtRPW8.2* genes in *N. benthamiana* leaves; (**d**) representation of DAB-positive stained clusters in tobacco leaves. Number of stained clusters was calculated per cm^2^ and repeated three times; (**e**) confirmed expression of fusion protein in *N. benthamiana* leaves. Error bars represent the standard deviation of three independent experiments. Columns with different letters represent significant difference according to Duncan’s multiple range test at *p* < 0.05.

**Figure 5 ijms-23-12588-f005:**
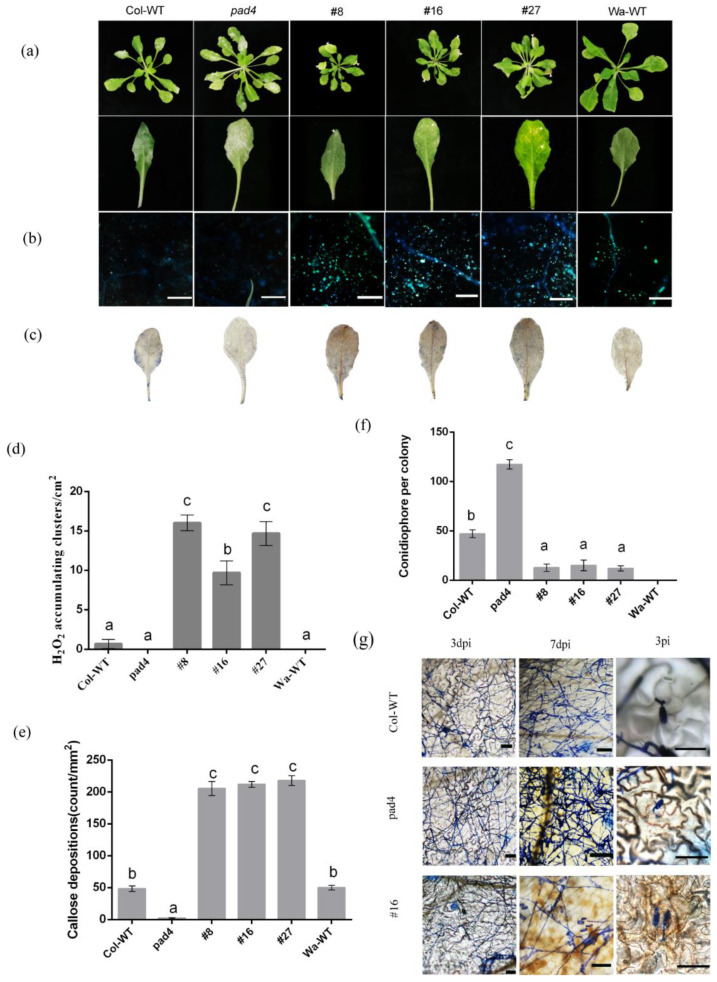
Overexpression of *HbRPW8-a* prevent powdery mildew colonization. (**a**) Over-expressing lines (#8, #16, #27), *pad4* mutant, Col-WT and Wa-WT plants were infected with powdery mildew. Col-WT, as a control, leaves were covered with hyphae; *pad4*, as a sensitive line, leaves were covered layer of hyphae; The over-expression line leaves covered a few hyphae; Wa-WT, as a disease resistant plant, with almost no hyphae on leaves. Plants were photographed 7 dpi. Plant cell death induced by *E. cichoracearum* colonies was highlighted by the arrow; (**b**) The content of callose depositions in the leaves of over-expression lines was the highest, followed by Wa-WT, Col-WT, and *pad4*. Callose deposition were stained with DAB at 48 hpi, and observed under microscope UV light; scale bar = 100 μm. (**c**) A large number of reactive oxygen species accumulation clusters appeared in the leaves of overexpression lines, accompanied by densely cell death; Col-WT leaves showed a few reactive oxygen species accumulation clusters and cell death; pad 4 and Wa WT showed slight cell necrosis. The accumulation of ROS and cell death in different plants at 3 dpi. ROS was visualized with DAB. Plant cell death were stained with trypan blue; (**d**) representation of DAB-positive stained clusters in over-expressing lines, *pad4*, Col-WT and Wa-WT leaves of per cm^2^. Data were shown as the means ± SD from three independent experiments and columns with different letters indicate significant difference (*p* < 0.05). (**e**) average numbers of callose deposits per 1 mm^2^ are displayed in over-expressing lines, *pad4*, Col-WT and Wa-WT leaves at 48 hpi. Data were shown as the means ± SD from three independent experiments and columns with different letters indicate significant difference (*p* < 0.05); (**f**) quantitative analysis of conidiophore formation on over-expressing lines, *pad4*, Col-WT and Wa-WT leaves at 7 dpi. Data were shown as the means ± SD from three independent experiments and columns with different letters indicate significant difference (*p* < 0.05); (**g**) fungal structures on leaves of Col-WT, *pad4* and #16 stained with trypan blue at 3 dpi and 7 dpi; From 3dpi to 7dpi, the number of hyphae on the leaves became more, especially on *pad4* leaves, the fungal structures was densely spread on the leaves. The fungal structures of Col-WT was less than that of *pad4*, but more than that of #16. And the epidermal cells of #16 were enriched in hydrogen peroxide surrounding germinating conidia, which was not find in Col-WT or *pad4* leaves; scale bar = 50 μm.

**Figure 6 ijms-23-12588-f006:**
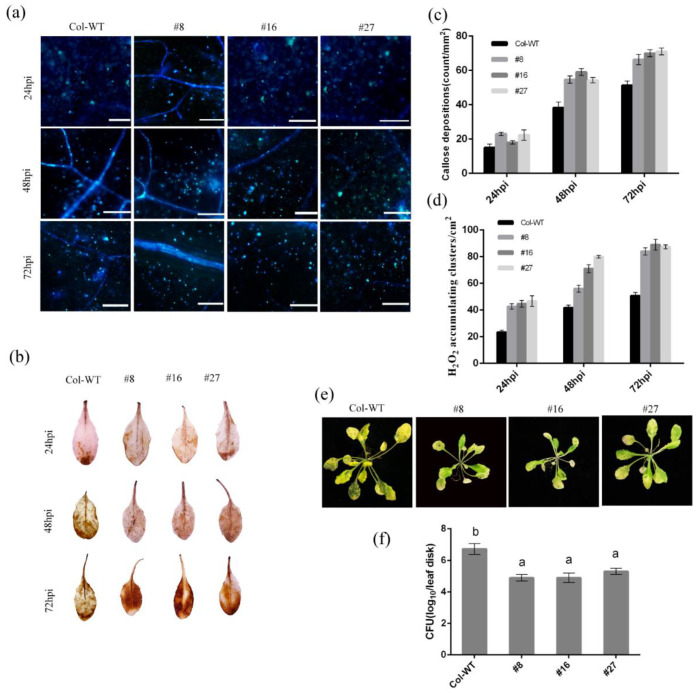
The responses of *HbRPW8-a* over-expressing lines and Col-WT to bacterial infection. (**a**,**b**), flg22 induced strong immune response in over-expressing lines (**a**) The callose deposition of Col-WT and over-expressing lines leaves induced by flg22 at 24, 48, 72 hpi; scale bar = 100 μm. (**b**) ROS accumulation in the overexpressing lines and Col-WT plants at 24, 48, 72 hpi; (**c**) average numbers of callose deposits per mm^2^ are displayed in Col-WT and over-expressing lines at 24, 48, 72 hpi; Data were shown as the means ± SD from three independent experiments (**d**) representation of DAB-positive stained clusters in Col-WT and over-expressing lines leaves. Number of stained clusters was calculated per cm^2^ and repeated three times; (**e**) phenotypes of Col-WT and overexpression lines infected with *Pst*DC3000 at 7 dpi; (**f**) bacterial growth in the leaves was determined at 3 dpi. Data were shown as the means ± SD from three independent experiments and columns with different letters indicate significant difference (*p* < 0.05).

**Figure 7 ijms-23-12588-f007:**
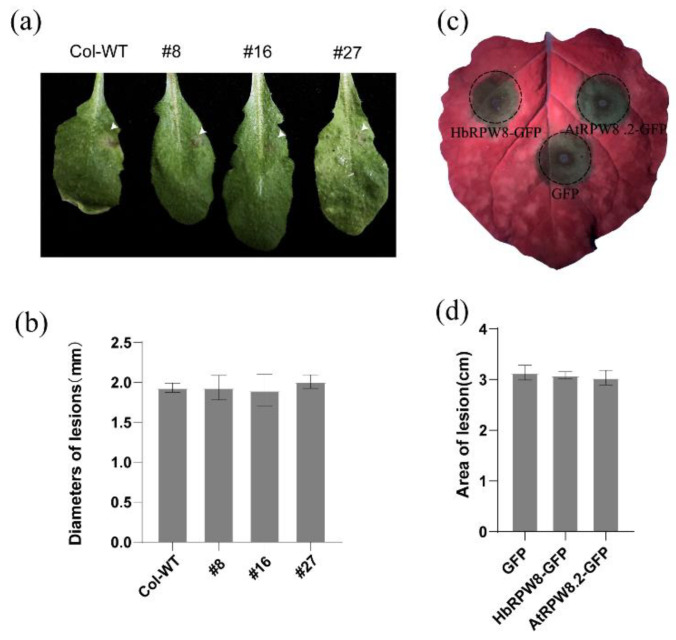
The responses of *HbRPW8-a* over-expressing lines and Col-WT to *P. capsici* or *B. cinerea* inoculation. (**a**) Macroscopic evaluation of symptoms 3 dpi with *B. cinerea* inoculation of Col-WT and transgenic lines; necrotic area was highlighted by the arrow; (**b**) and (**d**) measurement of lesion size at 3 dpi; Data were shown as the means ± SD from three independent experiments and no significant difference (*p* < 0.05). (**c**) macroscopic evaluation of symptoms at 3 dpi with *P. capsici* inoculation of tobacco leaf expressed with GFP, HbRPW8-a-GFP or AtRPW8.2-GFP.

**Figure 8 ijms-23-12588-f008:**
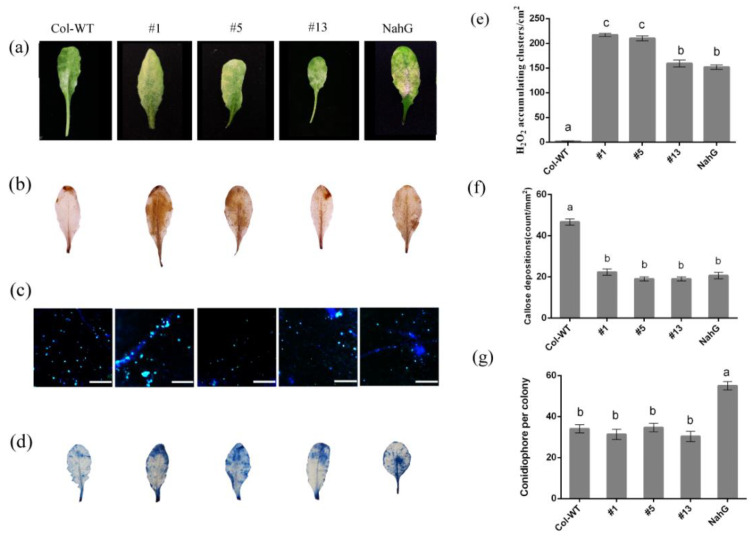
Phenotype of *HbRPW8-a* over-expressing lines in *NahG* background inoculated with *E. cichoracearum*. (**a**) Over-expressing lines (#1, #5, #13), *NahG*, Col-WT plants were infected with powdery mildew and photographed at 7 dpi; Col-WT as a control, its disease syptom was the same as that of transgenic lines. The *NahG* disease syptom was serious, with a large number of hyphae covered on leaves. (**b**) the accumulation of ROS and cell death in over-expressing lines, *NahG* and Col-WT leaves, photographed at 3 dpi; in *NahG* background, plant leaves with a large number of ROS clusters, but a few in Col-WT leaves (**c**) plant cell callose deposition were stained with DAB at 48 hpi in over-expressing lines, *NahG* and Col-WT leaves, and observed under microscope UV light; in Col-WT leaves, there were a large number of callose deposition, but a few in *NahG* background plant leaves, scale bar = 100 μm. (**d**) Severe cell death occured in *NahG* background plant leaves, photographed at 3 dpi; (**e**) representation of DAB-positive stained clusters in over-expressing lines, *NahG* and Col-WT leaves. Number of stained clusters was calculated per cm^2^; (**f**) average numbers of callose deposits per 1 mm^2^ are displayed in over-expressing lines, *NahG* and Col-WT leaves at 48 hpi. (**g**) quantitative analysis of conidiophore formation on over-expressing lines, *NahG* and Col-WT leaves at 7 dpi. All the data were shown as the means ± SD from three independent experiments and columns with different letters indicate significant difference (*p* < 0.05).

**Figure 9 ijms-23-12588-f009:**
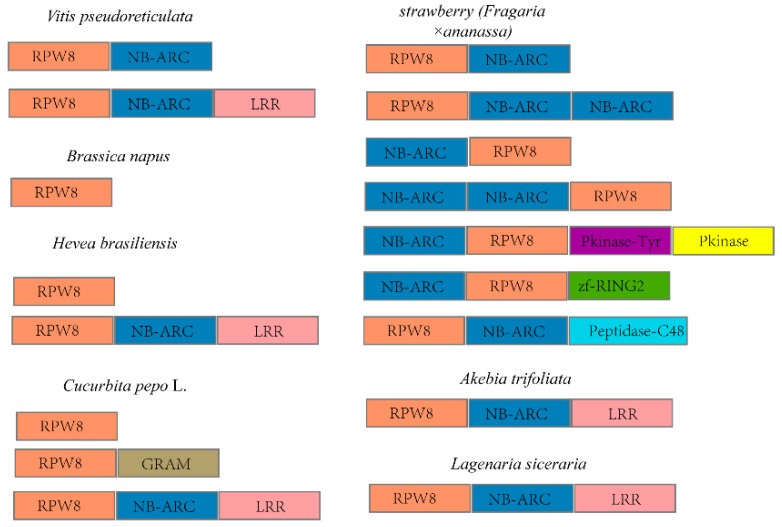
Various RPW8 proteins exist in plants. In *Brassica napus*, *Akebia trifoliata* and *Lagenaria siceraria (Molina) Standl.*, they only had one type of RPW8 fusion protein (RPW8 or RPW8-NR-ARC-LRR). In *V. pseudoreticulata* and *H. brasiliensis*, they had two types RPW8 fusion protein (RPW8/RPW8-NR-ARC, RPW8-NR-ARC-LRR). In *Cucurbita pepo* L., it had three types RPW8 fusion protein (RPW8, RPW8-GRAM, RPW8-NR-ARC-LRR). In *strawberries*, it had seven types RPW8 fusion protein (RPW8-NR-ARC, RPW8-(NB-ARC)_2_, NB-ARC-RPW8, (NB-ARC)_2_ CC RPW8, RPW8-NB-ARC-CC-Pkinase_Tyr- Pkinase, NB-ARC- RPW8-zf-RING_2_, RPW8-CC-NB-ARC-Peptidase_C48).

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
