# Peer review of "Ectopic Expression of HbRPW8-a from Hevea brasiliensis Improves Arabidopsis thaliana Resistance to Powdery Mildew Fungi (Erysiphe cichoracearum UCSC1)"

_ijms, 2022, doi:10.3390/ijms232012588_

Round 1

Reviewer 1 Report

Li et al. have identified and validated the function of an important RPW8 gene from rubber tree for its role in resistance against several bacterial and fungal pathogen. Overall, the manuscript represents a well-planned and well-executed study to validate the function of a putative candidate gene. The manuscript is well-written with appropriate data representation and interpretation. However, there are still some corrections to be addressed before accepting this manuscript for publication in this journal. Overall, I would like to recommend this manuscript for publication with “minor revision”.

Comment 1: Figure 1a is of very poor quality. It is impossible to read the presented information.

Comment 2: The scale bar is not visible in Figure 3, Figure 5g and Figure 6a.  

Comment 3: In Figure 9, provide the gene/protein accession numbers for various RPW8 genes in mentioned plants.

Comment 4: I would suggest including the data about raw gel image in supplementary file with proper defining the well details.  

Some minor corrections/ comments

Comment 1: Correct “A. thaliana” to “A. thaliana” in the legend of Figure 6d.

Comment 2: Correct “director” to “direct or” in line 6 of last paragraph of discussion.

Reviewer 2 Report

English should improve by a native person. The paper suffers from a poor English structure throughout and cannot be published or reviewed properly in the current format. The manuscript requires a thorough proofread by a native person whose first language is English. The instances of the problem are numerous and this reviewer cannot individually mention them. It is the responsibility of the author(s) to present their work in an acceptable format. Unless the paper is in a reasonable format, it should not have been submitted.

2.    The novelty of the study needs to be highlighted compare to other similar studies.

3.    Discussion is weak. The discussion needs enhancement with real explanations not only agreements and disagreements. Authors should improve it by the demonstration of biochemical/physiological causes of obtained results. Instead of just justifying results, results should be interpreted, explained to appropriately elaborate inferences. Discussion seems to be poor, didn't give good explanations of the results obtained. I think that it must be really improved. Where possible please discuss potential mechanisms behind your observations. You should also expand the links with prior publications in the area, but try to be careful to not over-reach. For the latter, you should highlight potential areas of future study.

4.    The scientific background of the topic is poor. In "Introduction" and "Discussion", the authors should cite recent references between 2016-2020 from JCR journals.

5. A detailed "Conclusion" should be provided to state the final result that the authors have reached. Please note you only need to place your conclusion and not keep putting results, because these have already been presented in the manuscript. 

6. Abstract is not well written; it is only a mere conscript of the study. Better would be to give some introduction followed by the gap in knowledge, hypothesis, general results and then conclusion. The abstract is the only part of the paper that the vast majority of readers see. Therefore, it is critically important for authors to ensure that their enthusiasm or bias does not mislead the reader.

Reviewer 3 Report

Manuscript entitled “Ectopic expression …….. UCSC1)” showed the cloning of five homologous HbRPW8 genes from rubber tree and expression profiling was performed. The study demonstrate the role of HbRPW8-a gene in plant resistant.

I have few concerns as follows:–

1.       If HbRPW8-b and HbRPW8-c had the identical DNA sequences, why they named differently? Might be you have isolated same homologues of gene and misunderstood the naming?

2.       How many homologous of AtRPW present in Arabidopsis plants? How they differ? There is possibility that you have cloned RPW gene with different length and confused with homologous.

3.       Have you used proof-read enzyme for cloning, if not why?

4.       Authors functionally validated this gene in Arabidopsis, is Arabidopsis is susceptible with powdery mildew fungi (Erysiphe cichoracearum)? If not, then how we can see the functional validation of this gene?

5.       Similar homologous gene is also present in Arabidopsis (AtRPW), how they kept WT as control?

6.       For functional validation, they have to use AtRPW mutant lines to get a clear control and characterization of gene.

Round 2

Reviewer 2 Report

Accepted as it stands

Reviewer 3 Report

Revision satisfactory